# Variation of sugar compounds in *Phoebe chekiangensis* seeds during natural desiccation

**Huangpan He** [ID]¹, **Handong Gao** [ID]¹*, **Xiaoming Xue**², **Jiahui Ren**¹, **Xueqi Chen**¹, **Ben Niu**¹

**1** College of Forestry and Grassland, College of Soil and Water Conservation, Nanjing Forestry University, Southern Tree Seed Inspection Center, National Forestry and Grassland Administration, Co-Innovation Center for Sustainable Forestry in Southern China, Nanjing, China, **2** College of Criminal Science and Technology, Nanjing Police University, Key Laboratory of Wildlife Evidence Technology of National Forestry and Grassland Administration, Nanjing, China

* gaohd@njfu.edu.cn

**Data Availability Statement:** All relevant data are within the paper and its Supporting Information files.

## Abstract

To investigate the role of sugar metabolism in desiccation-sensitive seeds, we performed a natural desiccation treatment on *Phoebe chekiangensis* seeds in a room and systematically analyzed the changes in seed germination, sugar compounds, malondialdehyde, and relative electrical conductivity during the seed desiccation. The results revealed that the initial moisture content of *P. chekiangensis* seed was very high (37.06%) and the seed was sensitive to desiccation, the germination percentage of the seed decreased to 5.33% when the seed was desiccated to 22.04% of moisture content, therefore, the seeds were considered recalcitrant. Based on the logistic model, we know that the moisture content of the seeds is 29.05% when the germination percentage drops to 50% and that it is desirable to keep the seed moisture content above 31.74% during ambient transportation. During seed desiccation, sucrose and trehalose contents exhibited increasing trends, and raffinose also increased during the late stage of desiccation, however, low levels of the non-reducing sugar accumulations may not prevent the loss of seed viability caused by desiccation. Glucose and fructose predominated among sugar compounds, and they showed a slight increase followed by a significant decrease. Their depletion may have contributed to the accumulation of sucrose and raffinose family oligosaccharides. Correlation analysis revealed a significant relationship between the accumulation of sucrose, trehalose, and soluble sugars, and the reduction in seed viability. Sucrose showed a significant negative correlation with glucose and fructose. Trehalose also exhibited the same pattern of correlation. These results provided additional data and theoretical support for understanding the mechanism of sugar metabolism in seed desiccation sensitivity.

## Introduction

*Phoebe chekiangensis* is a second-class protected tree species in China, a newly discovered endemic and endangered species in Tianmu Mountain of China [1], and its economic value is also high [2]. It provides high-quality wood that can be used for high-grade construction,

**Funding:** This study was supported by Postgraduate Research & Practice Innovation Program of Jiangsu Province (KYCX22_1110) (Huangpan He), which funded the cost of sampling and testing of samples.The funders had no role in study design, data collection and analysis, decision to publish, or preparation of the manuscript.

furniture, carvings, and precision molds [2]. It is also widely used in the production of plywood, lacquer, and wooden tires [2]. *P. chekiangensis* is distributed in Zhejiang, Fujian, Jiangxi, and southern Anhui Provinces in China, and grows mostly in evergreen broad-leaved forests in hilly valleys or red-soil slopes with an altitude of less than 1000 m, and it prefers a warm, moist climate and acidic or slightly acidic soil [3]. There are broad-leaved evergreen forests with *P. chekiangensis* as the dominant species in Yunqi and Li'an Temple in Hangzhou, while other areas are scattered [2,4]. Currently, the wild resources of *P. chekiangensis* have been seriously damaged and gradually exhausted due to its poor natural regeneration ability and destructive cutting [5]. *P. chekiangensis* mainly relies on sowing and seedling raising, and there is a lack of breeding and promotion of excellent strains [6]. Furthermore, the acceleration of global climate change, such as more frequent and prolonged droughts and rising temperatures, is a serious challenge for recalcitrant species that are highly dependent on specific habitats. Therefore, it is crucial to protect the preservation of *P. chekiangensis* germplasm resources. Assessing the desiccation tolerance of seeds serves as the foundation for germplasm conservation. Our pre-experiment found that *P. chekiangensis* seeds have higher moisture content (MC) after abscission and are sensitive to desiccation. In addition, there is dormancy in the seeds of *P. chekiangensis*, and variable temperature stratification can break the dormancy better [7,8]. Studying the desiccation sensitivity of the seeds can provide a theoretical basis and technical support for the long-term preservation of germplasm resources and biodiversity protection.

The majority of seed plants can produce desiccation-tolerant seeds (orthodox seeds), which have a remarkable ability to survive in extreme environmental conditions [9,10]. Orthodox seeds undergo maturation desiccation at a later stage of development, and the moisture content of the seeds is low after abscission and can generally be further dried to less than 5% moisture content without damage [11]. These seeds can be dried to less than 7% MC and stored at -18°C for a long time [12–14]. It is estimated that approximately 8% of seed plants worldwide exhibit seed desiccation sensitivity (recalcitrance), with this percentage increasing to 50% in tropical evergreen rainforests [12,15]. Recalcitrant seeds are highly susceptible to desiccation and freezing, and they quickly lose viability when stored under natural conditions, these seeds remain metabolically active throughout development and even after harvest [16].

Previous studies have indicated that seed acquires desiccation tolerance while accumulating large amounts of soluble sugars during development [17–20]. Among the soluble sugars, sucrose and raffinose family oligosaccharides (such as raffinose and stachyose) are considered to be involved in the formation of glass or have a protective effect on membrane phospholipids, or both [21]. Oligosaccharides will enhance the protective effect of sucrose by limiting crystallization and facilitating the vitrification of mature seeds [21,22]. Pukacka et al. [23] observed that during late development, desiccation-tolerant *Fagus sylvatica* L. seeds exhibited an increase in raffinose content and a decrease in the ratio of sucrose to raffinose, and there was a significant increase in the sucrose/(raffinose + stachyose) ratio and α-galactosidase activity in the seeds after 7 and 12 years of storage. Zhu et al. [24] discovered that subjecting the rice seeds to a slow-drying treatment at different developmental stages could induce sucrose accumulation. Trehalose is one of the most effective sugars for maintaining the structural and functional integrity of membranes and proteins at low water concentrations [25]. Coutinho et al. [26] found that among yeasts that naturally produce trehalose, mutant yeast strains with higher trehalose synthase activity had a greater ability to survive under drying and freezing conditions. Similarly, Bellaloui et al. [27] observed a significant increase in the concentration of stachyose, along with a significant decrease in the concentration of glucose, fructose, and sucrose, in Soybean seeds under water stress and high temperatures. Kristina and Sharon [28] found that the sucrose content of cotyledons of *Quercus alba* was increased in the later stages

of drying. The soluble sugar content was increased in recalcitrant seeds of *Quercus variabilis* [29] during drying.

In addition, Chandra and Keshavkant [30] reported a 1.6-19-fold increase in lipid peroxidation products in dehydrated *Madhuca latifolia* seeds, and they found an inverse relationship between water content and germination percentage. The drying-induced reduction in seed viability was associated with the accumulation of reactive oxygen species (ROS), which promote the loss of membrane integrity associated with lipid peroxidation. Liu et al. [31] reported an increase in ROS ($O_2^{\bullet-}$ and $H_2O_2$) and MDA content in germinating rice seeds under high temperatures and drought stress. Dresch et al. [32] found that drying may disrupt the integrity of the cell membrane system of *Campomanesia adamantium* seeds, increasing the concentration of exudates leached from the seeds, and leading to an increase in relative electrical conductivity.

To the best of our knowledge, current research on *P. chekiangensis* seeds has focused on seed dormancy and measures to break dormancy [7,8], as well as seed germination characteristics [33], while research on the response of *P. chekiangensis* seeds to desiccation has hardly been reported. Therefore, in this study, we systematically investigated the changes in seed germination, sugar compounds, malondialdehyde, and relative electrical conductivity during the natural drying of the seeds in a room, focusing on the role of sucrose, oligosaccharides, and hexoses (glucose and fructose) during seed desiccation, to provide additional data and theoretical support for the sugar metabolism mechanism of desiccation-sensitive seeds.

## Materials and methods

### Plant material and treatments

The seed maturation period of *P.chekiangensis* is about from late October to mid-November, and the seed dispersal period is about from late November to mid-December [34,35]. The seeds of *P. chekiangensis* were collected from three trees in Jiaguan Town, Qionglai City, Sichuan Province, China (31°17′31″N, 103°13′45″E), and the three trees were between 24 and 30 years old. The maturity of the seeds was determined by the change in color of the pericarp from green to blue-black. We picked the mature fruits in plastic bags on November 11, 2021, and brought them back to the laboratory immediately for processing. After returning to the laboratory, the peel was removed to obtain the seeds, and the cracked and broken seeds were removed by water selection. Next, the seed surface was dried in the shade, and the seeds with uniform size were selected for a natural desiccation test. The remaining seeds were stored in wet sand in a 4°C cold storage. The initial moisture content of *P. chekiangensis* seeds was measured using the low constant temperature oven drying method (103°C for 17 h) [36,37], and the thousand seed weight was measured using the hundred seeds method [38].

For the desiccation experiment, the seeds were spread flat in a circular frame on a shelf in the room for natural drying (temperature range: 15–21°C, relative humidity range: 30–53%). Additionally, six square porcelain plates were placed on the shelf, with each plate evenly spread with approximately 250 g of fresh seeds, and the actual weight of the seeds in these plates was recorded. According to the following Formula (1), the moisture content of the seed lot during desiccation was calculated by the weighing method [39].

Calculation formula of moisture content (MC) [39]:

$$R = \frac{M2 - M1 \times (1 - A)}{M2} \times 100\% \tag{1}$$

where: R is the seed moisture content, %; M1 is the mass of the undried seed, g; M2 is the mass of the dried seed, g; A is the initial moisture content of the undried seed, %.

Samples of the seeds were randomly taken under different drying treatments (37.06%, 33.99%, 30.63%, 28.11%, 25.09%, 22.04%, 19.08%, 16.04%, 13.05%, and 9.66%). Fresh undried (37.06%) seeds were used as the control group. A thermo-hygrometer was placed on the shelf to record the daily temperature and humidity in the room throughout the experiment. During the desiccation process, all experimental seeds were regularly mixed to ensure uniform drying. During each stage of drying (including fresh seeds), four replicates of 100 seeds each were taken for the determination of seed germination, and three replicates of 40 seeds each were taken for the determination of the remaining indicators. The seed drying rate index (k) was calculated by dividing the amount of water loss on a wet basis (%) by the corresponding elapsed time in hours [40].

### Determination of seed germination percentage

During each stage of drying (including the control), a total of 400 intact seeds with uniform size were randomly selected and placed on a wet cotton bed. Since the seeds of *P. chekiangensis* have dormancy [7,8], after drying treatments, all the seeds, including the control, were subjected to variable temperature stratification (15˚C 16 h / 25˚C 8 h, without light) for 60 days to break dormancy, and germination tests were carried out after dormancy release. The stratified seeds were taken out, washed with water, and then cultured in a constant temperature incubator at 30˚C with 24 hours of light. The germination of the seeds was observed and recorded daily. The seed germination standard was that the length of the radicle should be equal to the length of the seed. On the 30th day, the germination tests were finished, and the germination percentage of the seeds was calculated.

### Determination of soluble sugar and starch

Sample pre-treatment: we took the seed samples from the -80˚C refrigerator, removed the seed coat, and then put every 40 seeds (one repeat) into a low-temperature grinder to grind into powder, and mixed the powder of 40 seeds into a 50ml centrifuge tube and then placed it in liquid nitrogen for temporary preservation. After all the samples were treated in this way, the 27 (9 treatments, 3 replicates per treatment) centrifuge tubes filled with powder were placed in a refrigerator at -80˚C for testing.

The soluble sugar and starch were determined by the Anthrone colorimetric method referring to the experimental methods of Zhang [41] and Gao [42]. For each replicate, 0.5 g of seeds were weighed into 10 mL centrifuge tubes. Soluble sugars were extracted using 80% ethanol, while the remaining residue was used to extract starch with $9.2mol \cdot L^{-1}$ $HClO_4$ and $4.6mol \cdot L^{-1}$ $HClO_4$. 2 mL of soluble sugar extract was absorbed into a centrifuge tube, evaporated in a boiling water bath, then 10 mL of distilled water was added, and dissolved with sufficient stirring. The supernatant was obtained by centrifugation at $4000 \ r \cdot min^{-1}$ for 10 min. 2 mL of the supernatant was absorbed into the test tube and mixed with 5 mL of anthrone-sulfuric acid reagent in an ice-water bath. The reaction solution was mixed and then placed in a water bath at 100˚C for 10 min. After cooling to room temperature, the absorbance (OD) values of the reaction solution at 620 nm were read by a Beckman DU 800 UV-visible spectrophotometer (Beckman Coulter, Inc., Brea, CA, USA; the same hereafter). A standard curve was created using a glucose standard solution to calculate the soluble sugar content. Additionally, the starch extract was diluted 10 times, and the remaining steps were consistent with the determination of soluble sugar.

### Determination of sugar components

The determination of sugar components (including arabinose, fructose, galactose, glucose, inositol, sucrose, trehalose, raffinose, and stachyose) was carried out by referring to the

experimental method of Najah M. Al-Mhanna [43], and the qualitative identification and quantitative analysis of each compound were performed by gas chromatography-mass spectrometry (GCMS).

Extraction of the sample: Sample pre-treatment as above. 0.2 g of the sample was accurately weighed into a centrifuge tube, and 1 mL of petroleum ether was added. The resulting mixture was then centrifuged, and the supernatant was discarded to obtain the residue. Then 3 mL of 80% methanol was added into the residue for extraction, and this extraction process was repeated three times, resulting in a total of 9 mL of the extracted solution.

Derivatization: 1 mL of sample extracted solution was absorbed into a centrifuge tube, and 1.5 mL of 20 mg·mL$^{-1}$ methoxy aminopyridine hydrochloride solution was added, shaken at 650 r·min$^{-1}$ for 1.5 hours at 37°C. Then 2.5 mL of N, O-bis (trimethylsilyl) trifluoroacetamide (containing 1% trimethylchlorosilane) was added, and the mixture was shaken at 650 r·min$^{-1}$ for 1 hour at 70°C. Next, the mixture was taken out and left at room temperature for 30 minutes, centrifuged to obtain the supernatant, and then the supernatant was determined using the machine. Additionally, 1 mL of the standard solution was taken, and the same derivatization steps were followed as mentioned above.

GCMS operating conditions: the instrument was a gas chromatography triple quadrupole tandem mass spectrometer, model TSQ 9000 AEI (GCMSMS). HP-5 chromatographic column (30 m × 0.25 mm × 0.25 μm), injection volume was 1 μL, gasification chamber temperature was 280°C, split ratio was 100:1, carrier gas was helium (99.999%), and flow rate was 1 mL·min$^{-1}$. The initial temperature of the chromatographic column was 80°C for 1 min, and the temperature was increased to 240°C at a rate of 6°C·min$^{-1}$ for 15 min.

## Determination of relative electrical conductivity

The relative electrical conductivity of the seeds was determined with some modifications according to the method of Matthews et al. [44]. All seed samples were used as experimental determinations. The seeds were rinsed 3 times with deionized water, and the surface of the seeds was blotted with absorbent paper. Each tube was filled with 15 seeds, 25 mL of deionized water was added, and mixed well, and the initial conductivity (R0) of the mixture was measured using a conductivity meter (INESA, DDS, type 307). The sample mixture was then left at a constant temperature of 25°C for 24 hours to allow sufficient swelling, and the conductivity (R1) of the seed leachate was determined. Subsequently, the test tube with the leachate was sealed with a preservative film and placed in a boiling water bath for 40 minutes. After cooling to room temperature and shaking the leachate several times, the conductivity (R2) of the boiling seed leaching solution was measured. The relative electrical conductivity of the seeds, R (%), was calculated according to the following equation.

$$R(\%) = \frac{R1 - R0}{R2 - R0} \times 100$$

where: R0 is the initial leachate conductivity of the seeds (μs·cm$^{-1}$), R1 is the leachate conductivity of the seeds at 25°C (μs·cm$^{-1}$) and R2 is the leachate conductivity of the seeds after the boiling water bath (μs·cm$^{-1}$).

## Determination of malondialdehyde

The malondialdehyde (MDA) content was determined with some modifications according to the method of Chen et al. [45]. Sample pre-treatment as above. 0.5 g of seed sample was weighed into a mortar, and 8 mL of 5% trichloroacetic acid (TCA) was added for grinding, and then the homogenate obtained was centrifuged at 3000 r·min$^{-1}$ for 10 minutes. Next, 2 mL

of the supernatant was transferred to a test tube and mixed with 2 mL of 0.67% thiobarbituric acid (TBA). The mixture was then boiled in a water bath at 100˚C for 30 minutes. After cooling, the mixture was centrifuged again, and the resulting supernatant was used as the test solution. The absorbance (OD) values of the test solutions were measured at 450 nm, 532 nm, and 600 nm using the spectrophotometer.

## Statistical analysis

The normal distribution diagram and Shapiro-Wilk test were used to assess the normality of all data, while the Levene test was used to assess the homogeneity of variance. For data that did not meet normality or homogeneity of variance, a natural logarithm conversion was performed.

The relationship between seed drying time and moisture content was assessed using a quadratic function, while the relationship between moisture content and germination percentage was assessed using a Logistic regression model. One-way ANOVA was employed to determine the effects of different drying treatments on germination percentage, germination index, sugar compounds, relative electrical conductivity, and MDA. Duncan's multiple comparisons were conducted to identify significant differences between treatments ($p < 0.05$). Pearson correlation analysis was performed on the physiological and biochemical indexes, and the relationship between germination percentage and sugar compounds was further analyzed using linear regression analysis. Principal component analysis (PCA) was performed on experimental data. Statistical analysis and visualization of all data were conducted using SPSS 22 and Origin 21.

## Results

### Changes in moisture content and seed germination during desiccation

The initial moisture content of *P.chekiangensis* seeds was 37.06%, and the thousand seed weight was 286.52 g. The moisture content of *P. chekiangensis* seeds decreased with increasing drying time (Fig 1A). The seed drying rate was observed to be non-uniform under room temperature conditions, with a rate of 0.107%/h at the early stage of desiccation ($k_1$, between 0–84 h), 0.078%/h at the middle stage of desiccation ($k_2$, between 84–162 h), and 0.042%/h at the late stage of desiccation ($k_3$, between 162–455 h). This indicates that the drying rate exhibited a trend of initially being fast and then slowing down. As the moisture content decreased, the seed germination percentage decreased (Fig 1C). At the initial moisture content (37.06%), the seed germination percentage was 91.17% and did not change significantly when dried to 33.99% MC. However, when dried to 30.63% MC, the germination percentage decreased to 72.33%, indicating the seed viability began to decline significantly. The germination percentage further decreased to 36.50% when the seeds were dried to 28.11% MC, at which point most of the seeds had died. Subsequently, when the seeds were dried to 13.05% and 9.66% MC, all the seeds were dead. The change trends of the seed germination index (Fig 1D) and germination percentage were consistent. A logistic regression model was used to evaluate the relationship between seed moisture content and germination percentage (Fig 1B), and the $R^2$ and *p*-value indicated that the regression equation was a strong and statistically significant fit. According to this equation, the moisture content of *P. chekiangensis* seeds is 29.05% when the germination percentage drops to 50%. To ensure the high viability of the seed lot during short-term storage or transportation at room temperature, the moisture content should be maintained at 31.74% or higher. Below a moisture content of 13.55% MC, the seeds lose all germination power. Therefore, a moisture content range of 37.06–13.05% was used for the subsequent data analysis in this study.

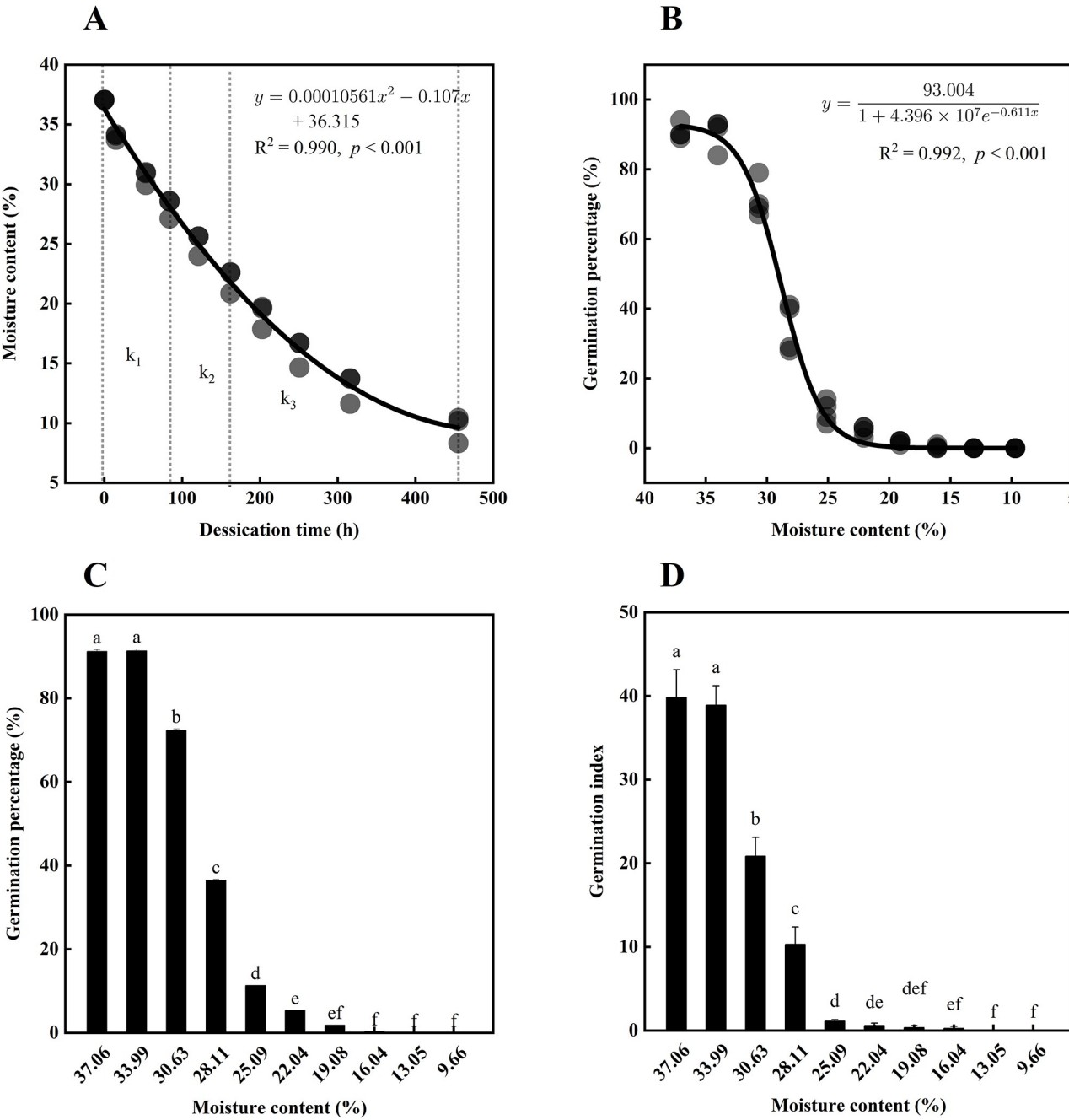

**Fig 1. Changes in moisture content and germination of *Phoebe chekiangensis* seeds during desiccation.** The quadratic function regression model was used for the relationship between drying time and moisture content (A), Logistic regression model was used for the relationship between moisture content and germination percentage (B). The influence of desiccation on germination percentage (C) and germination index (D) was analyzed by one-way ANOVA. Different lowercase letters represent significant differences at the $p < 0.05$ level according to the Duncan test. Data represent mean ± SE.

## Changes in sugar compounds during seed desiccation

The starch content increased and then decreased during drying, and it increased by 108.8% when dried to 30.63% MC (Fig 2A and S1 Table). However, when the seeds were dried to 25.09%, the starch content decreased by 34.76%. On the other hand, the soluble sugar content

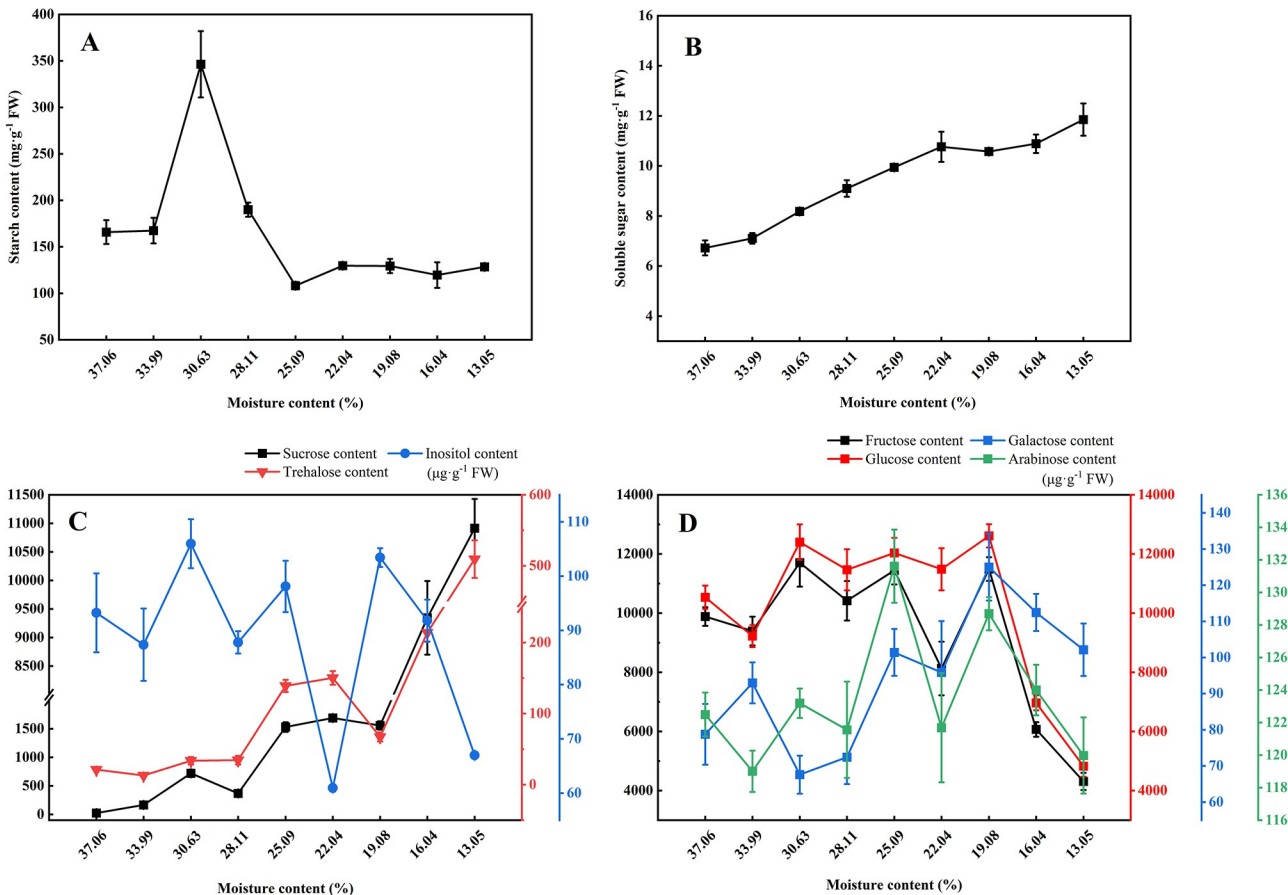

**Fig 2.** Effects of different moisture contents on starch (A), soluble sugar (B), and sugar components (C and D) during seed desiccation. Data represent mean ± SE.

exhibited a consistent increasing trend (Fig 2B and S1 Table). The soluble sugar content increased by 22% when dried to 30.63% MC and by 76% when dried to 13.05%.

Sucrose content significantly increased ($p < 0.05$) under drying treatment (Fig 2C and S1 Table). The sucrose content increased nearly six-fold when the seeds dried to 33.99% MC, while it increased about 447-fold at the late stage of drying (13.05% MC). Similarly, the trehalose content exhibited a fluctuating increase during drying. The trehalose content started to significantly increase at 30.63% of MC and continued to rise as drying progressed. Ultimately, the trehalose content increased about 23-fold when the seeds dried to 13.05% MC. During desiccation, there was a noticeable pattern in the inositol content. Inositol content was increased by 13.67% when the seeds were dried to 30.63% MC. However, as the seeds were further dried to 22.04% and 13.05% MC, the inositol content decreased by 34.65% and 28.17%, respectively. The glucose content showed a trend of increasing and then decreasing during seed desiccation (Fig 2D and S1 Table). Notably, the increase in glucose content was not significant when the seeds were dried to the range of 30.63% to 19.08% moisture content. However, a significant decrease ($p < 0.05$) in glucose content was observed when the seeds were dried to 16.04%. The trend of fructose during seed desiccation likely followed a similar pattern to that of glucose. The galactose content exhibited a fluctuating increasing trend as the moisture content decreased. When the seeds were dried to 25.09% MC, galactose content increased by 28.75%, while when the seeds were dried to 19.08% MC, galactose content increased by 58.72%. On the

other hand, the arabinose content showed an increase followed by a decrease during drying. The arabinose content significantly increased ($p < 0.05$) when the seeds were dried to 25.09% MC, while no significant change was observed under other drying treatments.

We found that the specific non-reducing sugar content (the sum of sucrose, trehalose, and inositol content) increased during drying, with an approximately 82-fold increase in the non-reducing sugar content at 13.05% MC (Fig 3A). In contrast, the ratio of (glucose + fructose + arabinose + galactose) to (sucrose + trehalose + inositol) (RS/NRS) was decreased during drying, with the ratio decreasing by almost 1-fold at 13.05% MC (Fig 3C). Among the three

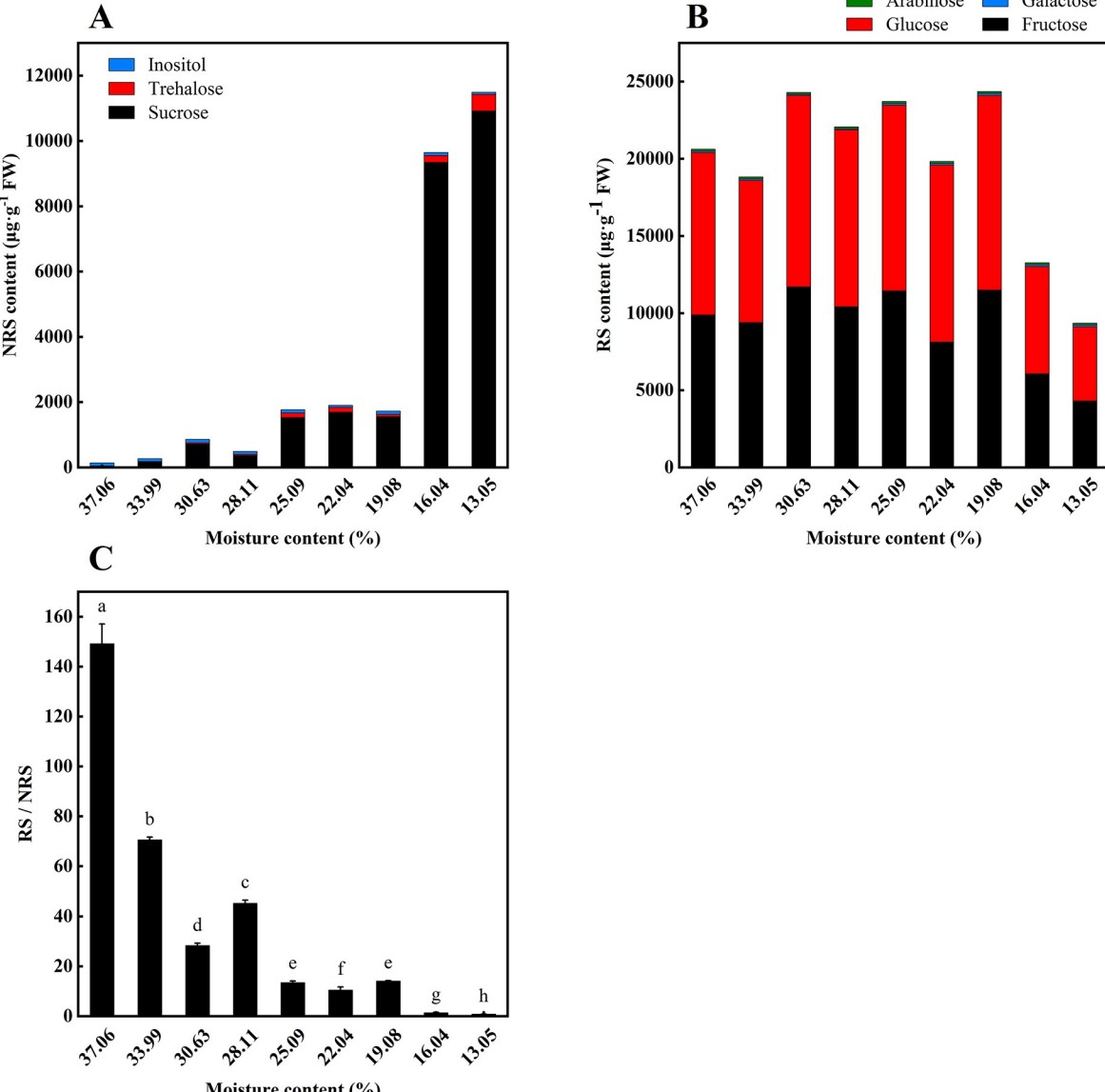

**Fig 3.** Effects of different moisture contents on (A) non-reducing sugar (NRS), (B) reducing sugar (RS), and (C) reducing sugar/non-reducing sugar (RS/NRS) during seed desiccation. The non-reducing sugar content refers to the combined amount of sucrose, trehalose, and inositol, whereas the reducing sugar content refers to the combined amount of fructose, glucose, galactose, and arabinose. RS/NRS refers to the ratio of (glucose + fructose + arabinose + galactose) to (sucrose + trehalose + inositol). One-way ANOVA analysis was performed on the data. Different lowercase letters represent significant differences at the $p < 0.05$ level according to the Duncan test. Data represent mean ± SE.

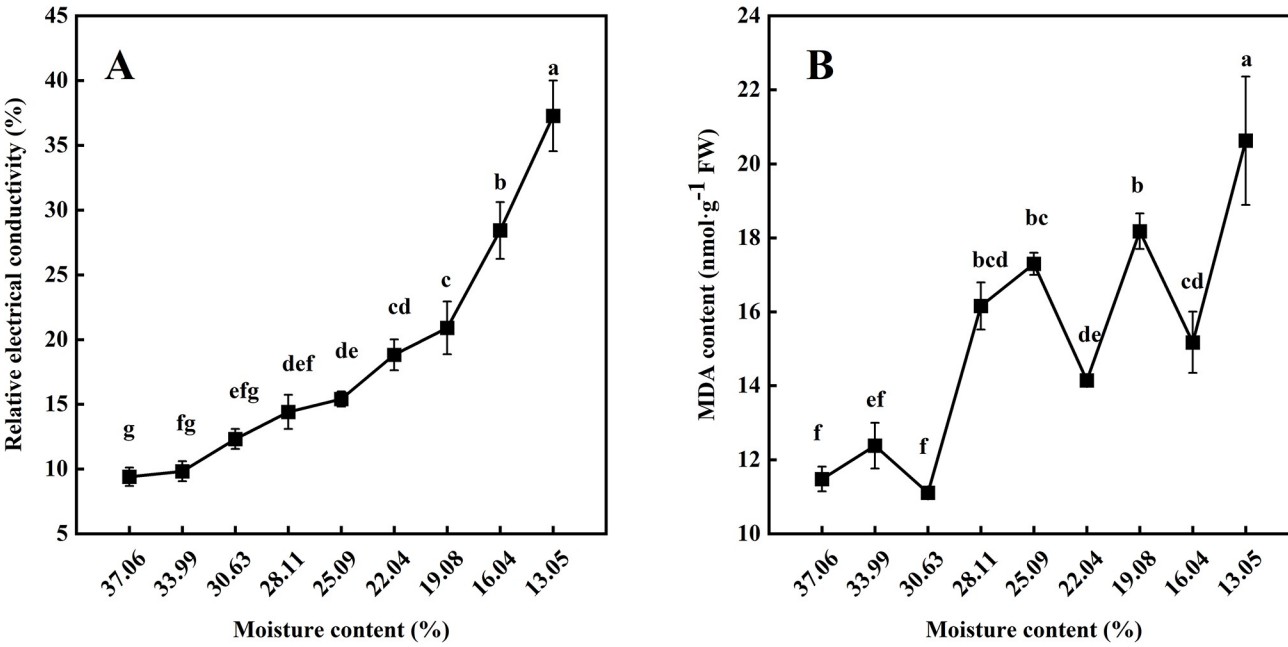

**Fig 4.** Effects of desiccation on relative electrical conductivity (A) and MDA (B). One-way ANOVA analysis was performed on the data. Different lowercase letters represent significant differences at the $p < 0.05$ level according to the Duncan test. Data represent mean ± SE.

non-reducing sugars (sucrose, trehalose, and inositol), sucrose was the most abundant, while glucose and fructose dominated among the four reducing sugars (Fig 3A and 3B).

## Changes in relative electrical conductivity and MDA during desiccation

As the seed moisture content decreased, the relative electrical conductivity (RC) tended to increase (Fig 4A), while the malondialdehyde (MDA) content showed a fluctuating upward trend (Fig 4B). Notably, both RC and MDA contents increased significantly ($p < 0.05$) when the seeds were dried to 28.11% MC, indicating that at this stage the seeds began to suffer damage. The seed germination percentage also significantly decreased to 36.50% at this time. Furthermore, when the seeds were dried to 13.05% MC, the relative electrical conductivity and MDA increased by 296.58% and 79.69%, respectively.

## Correlation and principal component analysis

During seed desiccation, the germination percentage showed a significant positive correlation ($p < 0.05$) with the germination index, starch, fructose, and RS/NRS (Fig 5). On the other hand, it was negatively correlated ($p < 0.05$) with soluble sugars, galactose, sucrose, trehalose, relative electrical conductivity (RC), and MDA. Soluble sugars exhibited a significant positive correlation with galactose, sucrose, and trehalose ($p < 0.05$), but a negative correlation with fructose, inositol, and RS/NRS ($p < 0.05$). Furthermore, fructose showed a significant positive correlation with glucose (r = 0.95, $p < 0.05$). Sucrose exhibited significant negative correlations with both fructose and glucose, with correlation coefficients of -0.58 and -0.50, respectively ($p < 0.05$). Additionally, sucrose showed a significant positive correlation with trehalose, with a correlation coefficient of 0.88 ($p < 0.05$). Glucose was positively and significantly correlated with inositol ($p < 0.05$), but negatively correlated with sucrose and trehalose ($p < 0.05$).

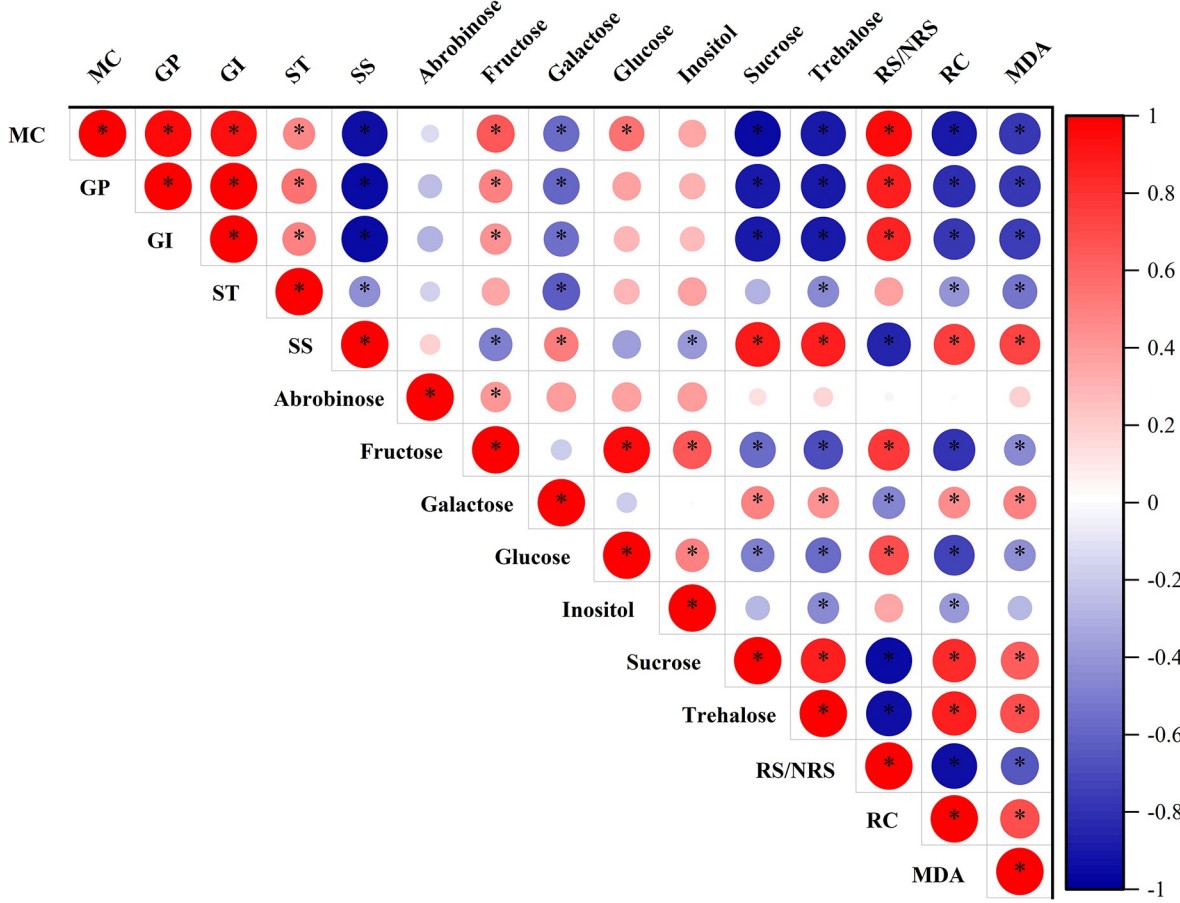

**Fig 5. Correlation analysis between physiological and biochemical indexes during seed desiccation.** The size of the circles in the diagram represents the level of correlation (r), with bigger circles indicating a higher correlation. "*" indicate significant differences at $p < 0.05$. MC: Moisture content, GP: Germination percentage, GI: Germination index, ST: Starch, SS: Soluble sugar, NRS: The combined amount of sucrose, trehalose, and inositol, RS: The combined amount of fructose, glucose, galactose, and arabinose, RS/NRS: The ratio of the two above, RC: Relative electrical conductivity, MDA: Malondialdehyde.

Fructose showed a significant positive correlation with arabinose and inositol ($p < 0.05$), and a significant negative correlation with trehalose ($p < 0.05$). Trehalose was found to have a significant negative correlation with starch, fructose, and inositol ($p < 0.05$). RC exhibited a significant positive correlation with soluble sugars, galactose, sucrose, and trehalose ($p < 0.05$), and a negative correlation with starch, fructose, glucose, inositol, and RS/NRS ($p < 0.05$). Additionally, RC showed a significant positive correlation with MDA (r = 0.69, $p < 0.05$). We continued to explore the relationship between germination percentage and sugar compounds by linear regression analysis (S1 Fig). It was observed that the germination percentage was better fitted with soluble sugar, sucrose, and trehalose with $R^2$ of 0.891, 0.741, and 0.758, respectively. All the fitted equations mentioned above were found to be significant.

The results of the principal component analysis revealed that the germination percentage had a major impact on principal component 1 (Prin1), followed by RS/NRS, germination index, and trehalose, which showed high loadings of these variables in Prin1 (Fig 6). Starch and galactose had a major impact on principal component 2 (Prin2), as evidenced by the high loadings of the two variables in Prin2. The first two principal components accounted for 71.3% and 11.1% of the observed variance, respectively (totaling 82.40%).

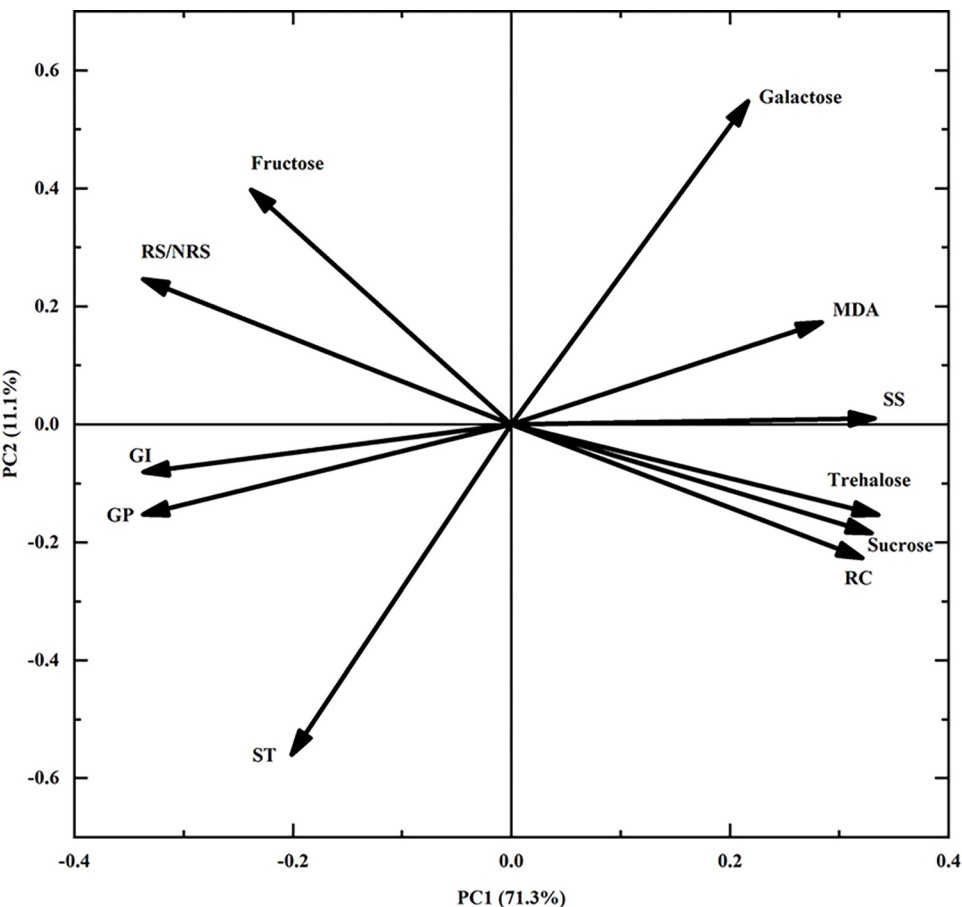

**Fig 6. Principal component analysis between physiological and biochemical indexes during seed desiccation.** GP: Germination percentage, GI: Germination index, ST: Starch, SS: Soluble sugar, NRS: The combined amount of sucrose, trehalose, and inositol, RS: The combined amount of fructose, glucose, galactose, and arabinose, RS/NRS: The ratio of the two above, RC: Relative electrical conductivity, MDA: Malondialdehyde.

## Discussion

### Desiccation sensitivity of *P. chekiangensis* seeds

The moisture content of recalcitrant seeds is high before and after harvest, accompanied by a state of high metabolism. The moisture content of the seeds of *P. chekiangensis* was also higher (37.06%) when they were matured and shed, and the seeds are sensitive to desiccation, seed germination percentage decreased to 5.33% when the moisture content decreased to 22.04% (Fig 1C), therefore, the seeds of *P. chekiangensis* are recalcitrant. The seeds drying rate at room temperature was uniform, which was consistent with the results of Vieira et al. [40] and Karin et al. [46]. In addition, differences in sensitivity to desiccation were observed among different species. Tchokponhoué et al. [47] found that the seeds of *S. dulcificum* had an initial moisture content of 36.6% (99% of viability). However, when the seeds were dried to 20%, more than half of their viability was lost, and when dried to 9.5%, all the seeds died. Varghese et al. [48] discovered that mahua (*Madhuca indica* J.F. Gruel.) seeds had a high initial MC (53%) with 100% of seed viability. However, after being dried for 29 days, reaching a moisture content below 16.8%, the seed germination percentage decreased by almost 90%.

## Response of sugars as the protective substance to seed desiccation

Previous studies have shown a correlation between sugar accumulation and seed desiccation tolerance [49–52]. Mechanisms to protect against desiccation damage may exist in both the vegetative organs and seeds of desiccation-tolerant plants, such as the accumulation of sucrose and raffinose family oligosaccharides (raffinose and stachyose), which help stabilize membranes, proteins, and the cytoplasmic glass matrix [53]. How do the sugars of recalcitrant seeds respond to desiccation? During desiccation, the contents of sucrose, trehalose, and soluble sugar in *P. chekiangensis* seeds increased significantly, and the starch content decreased. This is consistent with the findings of desiccation-sensitive *Inga vera* seeds [54]. Hydrolysis of starch may contribute to the increase in sugars. Moreover, the increase in soluble sugars observed in the seeds of *P. chekiangensis* may also be a result of reduced seed respiration rate, which reduces sugar consumption. Correlation analysis revealed that the accumulation of sucrose and trehalose was associated with reduced seed viability, indicating that the accumulation of the sugars acts as a protective substance in the seed, but low levels of sugar accumulation still do not prevent the loss of seed viability. Notably, it has also been argued that the impact of sugar on seed desiccation tolerance is not isolated but rather connected to both the developmental stage of the seed and the synergistic effects of abscisic acid and proteins [18].

The supply of hexoses (glucose and fructose) during desiccation is an important step in the regulation of sugar biosynthesis [55]. In our desiccation experiments, we observed that fructose and glucose predominated among soluble carbohydrates, and they showed a slight increase in the middle of desiccation but significantly decreased towards the late desiccation stage, which is in general agreement with the results of Ghasempour et al. [56] and Whittaker et al. [55]. We speculate that the slight increase in glucose and fructose in the middle of desiccation may be attributed to a higher rate of starch decomposition compared to sucrose synthesis at this stage. In addition, the transient accumulation of these sugars in the middle of desiccation contributes to osmoregulation [56]. We also found that the raffinose content of *P. chekiangensis* seeds increased during the later stages of desiccation (S1 Table), in contrast to the results for desiccation-sensitive *Inga vera* seeds [54]. The accumulation of high levels of hexoses and low levels of sucrose and oligosaccharides may be a characteristic of some desiccation-sensitive seeds.

## Response of MDA and relative electrical conductivity to desiccation

When plants were exposed to environmental stress, cell membranes were the first to be damaged, resulting in membrane lipid peroxidation, spillage of intracellular lysis products, and accumulation of malondialdehyde [57]. The content of MDA can indicate the extent of cell membrane peroxidation and cell damage [58]. During desiccation, the MDA content of *P. chekiangensis* seeds showed a fluctuating upward trend. A significant increase in MDA content was observed when the seeds were dried to 28.11% MC. This indicates that the seeds have experienced severe oxidative damage, resulting in a loss of more than half of their viability. Furthermore, we observed a correlation between MDA accumulation and reduction in seed viability. Previous studies have also confirmed that the decline in the viability of recalcitrant seeds is closely related to the accumulation of membrane lipid peroxides, such as *Camellia sinensis* [59], *Quercus robur* [60], *Trichilia connaroides* [61] and *Ginkgo biloba* [39]. Under abiotic stress conditions, loss of seed viability is linked to the disruption of cell membrane structure and integrity, leading to an increase in relative electrical conductivity [39]. We found that the relative electrical conductivity of *P. chekiangensis* seeds kept increasing during desiccation. A similar trend was also observed during the drying of *Campomanesia adamantium* seeds [32]. In conclusion, both MDA and relative electrical conductivity can better characterize the damage to the membrane system induced by seed desiccation.

## Conclusions

In this study, we analyzed the effects of different drying treatments on seed germination, sugar compounds, MDA, and relative electrical conductivity in *P. chekiangensis* seeds. The initial moisture content of the seeds was high (37.06%) and the seeds were sensitive to desiccation, for example, when the seeds were dried to 22.04% MC, they lost 94% of their viability, therefore, the seeds of *P. chekiangensis* were considered to be recalcitrant. According to the logistic model, we know that when seed germination drops to 50%, the moisture content is 29.05%, and the MC should ideally be kept above 31.74% when the seed lot is transported at room temperature. The contents of sucrose and trehalose tended to increase during seed desiccation. Correlation analyses showed that the reduction in seed viability was significantly correlated with the accumulation of sucrose, trehalose, and soluble sugars, suggesting that the sugar acts as a protective substance against desiccation, but low levels of the sugar accumulation were still unable to inhibit the loss of seed viability induced by desiccation. We observed that among the sugar compounds, glucose and fructose were predominant, with a slight increase in their content in mid-desiccation and a significant decrease in late desiccation. The accumulation of glucose and fructose in the middle stage of desiccation is beneficial for osmoregulation. Both MDA and relative electrical conductivity showed an increasing trend during seed desiccation, indicating potential damage to the membrane system induced by seed drying.

## Supporting information

**S1 Table. Changes of soluble carbohydrates in seeds of *Phoebe chekiangensis* during desiccation.** One-way ANOVA analysis was used for the data. Different lowercase letters represent significant differences at the $p < 0.05$ level according to the Duncan test. Data represent mean ± SE. "-" means the data is not detected.
(DOCX)

**S1 Fig. Linear regression analysis between germination percentage and some sugar compounds.** The regression line is represented by the black line. GP: Germination percentage, SS: Soluble sugar.
(DOCX)

## Acknowledgments

We would like to thank Mingzhu Wang, Wen Gu, Yuan Zheng, and Hao Li for their help in handling the experimental materials.

## Author Contributions

**Conceptualization:** Huangpan He.

**Data curation:** Huangpan He.

**Investigation:** Huangpan He, Jiahui Ren, Xueqi Chen, Ben Niu.

**Methodology:** Huangpan He.

**Writing – original draft:** Huangpan He.

**Writing – review & editing:** Handong Gao, Xiaoming Xue.

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
