## [Decision Letter · Decision Letter 0]

20 Nov 2023

PONE-D-23-28564Variation of soluble carbohydrates in Phoebe chekiangensis seeds during natural desiccationPLOS ONE

Dear Dr. Gao,

Thank you for submitting your manuscript to PLOS ONE. After careful consideration, we feel that it has merit but does not fully meet PLOS ONE’s publication criteria as it currently stands. Therefore, we invite you to submit a revised version of the manuscript that addresses the points raised during the review process.

**Variation of soluble carbohydrates in Phoebe chekiangensis seeds during natural desiccation**

The research is interesting. However, some points are there which must be cleared as mentioned by the reviewers. Discussion part needs refinements. Conclusions should be more precise, meaningful and should be based on data obtained. The overall English grammar of the manuscript needs to be improved. The manuscript can be accepted after these changes and those suggested by the reviewers.

We look forward to receiving your revised manuscript.

Kind regards,

Meenakshi Thakur, Ph.D.

Academic Editor

PLOS ONE

Journal Requirements:

3. Please include your figures as part of your main manuscript and remove the individual files. Please note that supplementary tables (should remain/ be uploaded) as separate "supporting information" files".

4. We are unable to open your Supporting Information file [Figs1-7.tif.7z]. Please kindly revise as necessary and re-upload.

Additional Editor Comments:

Variation of soluble carbohydrates in Phoebe chekiangensis seeds during natural desiccation

The research is interesting. However, some points are there which must be cleared as mentioned by the reviewers. Discussion part needs refinements. Conclusions should be more precise, meaningful and should be based on data obtained. The overall English grammar of the manuscript needs to be improved. The manuscript can be accepted after these changes and those suggested by the reviewers.

Reviewers' comments:

Reviewer's Responses to Questions

**Comments to the Author**

1. Is the manuscript technically sound, and do the data support the conclusions?

Reviewer #1: Yes

Reviewer #2: Partly

Reviewer #3: Yes

2. Has the statistical analysis been performed appropriately and rigorously? 

Reviewer #1: No

Reviewer #2: Yes

Reviewer #3: Yes

3. Have the authors made all data underlying the findings in their manuscript fully available?

Reviewer #1: Yes

Reviewer #2: Yes

Reviewer #3: Yes

4. Is the manuscript presented in an intelligible fashion and written in standard English?

Reviewer #1: No

Reviewer #2: Yes

Reviewer #3: Yes

5. Review Comments to the Author

Reviewer #1: I have to admit that I recently contacted one of the author’s departments for some collaboration and/or other possibilities; although it did not materialize, I have known one of the authors of this study. However, this does not exert a conflict of interest in reviewing this paper. I have reviewed this as I would review other papers.

I have an exceptional interest in Lauraceae, and several new species discovered in the 21st-century display that many species of this family can produce desiccation-sensitive seeds. A review article previously described understanding desiccation-sensitivity in Lauraceae is complex and requires careful consideration (Jaganathan et al., 2019). In this sense, this manuscript is fascinating. However, the manuscript needs a lot of improvement. Please see below where I cannot ascertain what the authors were trying to say.

The title can be improved. ‘Variation of soluble carbohydrates’ looks a little vague. Perhaps something to reflect the overall results would be good.

The writing can be a lot tighter. For example, the first two sentences of the introduction can be combined, and statements like these need reference.

Introduction

Can the authors think about giving a range of occurrences for this species?

L43- it is unclear what you mean by breeding technology.

L48- rising temperatures and more prolonged droughts are weather anomalies;- therefore, it is unclear what you mean here.

L49- didn’t you say this a few lines before?

L51- the high moisture content was observed in this study or previous studies. If the former, you should better put them in results.

L 52- You have a tendency to repeat the same information using different words. Please combine all these and write concisely.

L 58- order can be improved. First, talk about desiccation during development, then abscission.

L 60- I think storing at -10C is rare; the seeds are usually stored at -18C.

L67- 10% is too low; most recalcitrant seeds die above 10% MC.

L68- you say previous studies but cited only one reference.

L 69- I think you tend to say important and they are, why not delete ‘are important, and they’

L73- how?

L75- what do you mean by acquire?

L77- 106: The paragraph appears too long. I think you could condense the information. Similarly, these are for orthodox seeds. Is there any evidence showing soluble sugar accumulation is different in recalcitrant seeds? If so, you would have to discuss that.

L107- Perhaps you need to specifically state the objectives of the study here. It looks like you have only investigated the soluble carbohydrates in this study.

Materials and methods

L121- the collection time (maturity time) indicates the seeds are naturally dispersed before winter. Last year, when I visited Jiangsu, it was early December, and winter had begun already. It would be good to include more details about dispersal.

L126- for how long?

L 126- MC results move to results. However, what do you mean by the low-constant oven drying method? Please specify the temperature and time.

L 128- seed weight results to the results section. What is the 100 seeds method? Please include a reference; if not, describe how this was done.

L133- following formula? I don’t see any formula following, but if you mean the equation given on L149- use the number and refer accordingly.

L 136- I don’t think these were target moisture contents. I would assume you dried the seeds and periodically removed them; if so, why not give time instead of moisture content? I also think ‘portion’ is not a good usage in science. Two seeds can be a portion, and all the seeds except two can also be a portion.

L138-148: the order or the information should be tweaked for clarity.

L 155- did any studies show these seeds have dormancy, or is it your assumption? If this is your unpublished result, you should probably include this in the introduction.

L 156-160- a lot going on. Did you use 15/25C for the dormancy break? Why did you choose this temperature range? It is also unclear why 24-hour light was chosen for germination.

For sections 2.3 and 2.4, you mentioned the methodology but did not give any details about the materials. It is seeds, but what treatments were done to them is not given. This information is crucial.

L 217- Control seeds?

L 233- what is MDA?

L 246- so you did both square root and log transformation? Whilst which transformation is better could be a subject of unresolved debate, all researchers would agree that a consistent transformation should be followed.

Results

L 272- after drying, the viability loss occurred. However, Jaganathan et al. (2019) said the seeds can become dormant during drying. Indeed, you mentioned something about dormancy but did not test this.

L 281- what is semi-lethal?

L 295- Perhaps a better approach could include such analysis during seed development, at least from the maturation drying stage, i.e., when the pericarp colour starts to change.

Discussion

L 416- For P. chekiangensis seeds. What is this- this statement is incomplete.

L420- classifying them as recalcitrant seeds- there is something wrong with the whole sentence and expression.

L 434- I also think such variation might be due to family-level variation. Therefore, if possible, can you compare the seed coat structure and size of all these seeds in relation to the species studied here?

L442-462- The discussion of the resurrection plant is not necessary, in my opinion. You merely have to state this is the case in vegetative tissues and focus your attention on seeds.

L 475- From what is presented here, I don’t see any methods for evaluating the raffinose or other sugar during maturation.

The authors also speculate too much in the discussion and often mix desiccation-tolerant species results with the present study. It would be good if the authors toned down their arguments and placed more in the context of their results.

Minor comments:

Figure orders should be adjusted in the manuscript.

References

Jaganathan, G.K., Li, J., Yang, Y., Han, Y. and Liu, B. (2019) Complexities in identifying seed storage behavior of hard seed-coated species: a special focus on Lauraceae. Botany Letters. 166, 70-79.

Reviewer #2: The authors provided a precise study of the natural drying of seeds of endangered Chinese species Phoebe chekiangensis. Seeds seemed to be recalcitrant as most of them usually do not survive moisture content lower than 29%. The analysis of carbohydrate profile, content of starch and MDA, relative electrical conductivity and germination tests with precise statistical evaluation of obtained data would bring an interesting conclusion that the accumulation of reducing sugars fructose and glucose instead of non-reducing sugars sucrose and RFO (stachyose were not detected at all). Unfortunately,these data are not fully reliable, if you measured dead seeds (as you mentioned most of the seeds were dead at the moisture content below 28%) and if you calculate all values to fresh weight instead of dry weight. This is very important if the main changing parameter is moisture content.

In the description of methods, I would recommend to join into one chapter these chapters: 2.4, 2.4.1., 2.4.2. and 2.4.3. The last one is written like a protocol in the student notebook, please try to restyle into phrases. Chapter 2.5. could be divided into two chapters and not two subchapters.

Last but not least, I would recommend to submit this manuscript to some journal more releavnt to the topic and used methods, e.g., Seed Science and Technology.

Reviewer #3: The manuscript is well written but still have some grammatical errors such as: L 352. during this period, the seeds experienced severely damaged due to drying, with or L 416. The manuscript needs to be revised for such errors.

L 133. The formula should be provided at the end of this sentence.

L 311. % trehlose content increase (23.03) under drying seemed to be wrong. See Fig 2C

L 332-338. Concluding remarks (based on Fig 3) should be more precise and should be based on data obtained. Modify these sentences to draw more meaningful conclusions. Also check Fig 3C. It needs more clarity especially w.r.t. RS/NRS. Is it ratio of RS to NRS or something else?

L 389 – L 395. The relevance of correlation of germination with different parameters can be made more meaningful if glucose, fructose, galactose etc are also compared with germination. The Fig 6 failed to present meaningful information as it did not include data on above mentioned parameters. Moreover, there is no mention of positive and negative correlation. The Fig 5 and Fig 6 are almost similar and hence, one of the figures may be included as supplementary figure.

Similarly, for Fig 7, only those parameters that are significantly correlated to germination should be included for more meaningful presentation.

Discussion and conclusion parts also need refinements especially with respect to resurrection plants and recalcitrant plants. If sugar patterns are similar in both under drying, the present study apparently failed in its objective of understanding mechanisms leading to reduced germination under drying in the recalcitrant plant.

To draw meaningful conclusions, the seeds of P. chekiangensis must be rehydrated following drying to see if those resume physiological activities including germination following desiccation or not. Appropriate tables and figures related to this experiment may then be added to the manuscript.

6. PLOS authors have the option to publish the peer review history of their article (what does this mean?). If published, this will include your full peer review and any attached files.

Reviewer #1: **Yes: **Ganesh K. Jaganathan

Reviewer #2: No

Reviewer #3: **Yes: **Kamal Dev Sharma

---

## [Author Response · Author response to Decision Letter 0]

4 Jan 2024

Thank you very much for your efforts on our manuscripts, and we have responded to your constructive comments one by one. We hope our reply can make you satisfied, thank you again. We have uploaded a rebuttal letter named 'Response to reviewers '.

---

## [Editor Report · Decision Letter 1]

14 Feb 2024

Variation of sugar compounds in Phoebe chekiangensis seeds during natural desiccation

PONE-D-23-28564R1

Dear Dr. Handong Gao,

We’re pleased to inform you that your manuscript has been judged scientifically suitable for publication and will be formally accepted for publication once it meets all outstanding technical requirements.

Kind regards,

Meenakshi Thakur, Ph.D.

Academic Editor

PLOS ONE

Additional Editor Comments (optional):

Most of the comments of the reviewers have been justified by the authors. Hence, I am pleased to inform you that the manuscript can be accepted now for publication.
---

## [Editor Report · Acceptance letter]

28 Feb 2024

PONE-D-23-28564R1 

PLOS ONE

Dear Dr. Gao, 

I'm pleased to inform you that your manuscript has been deemed suitable for publication in PLOS ONE. Congratulations! Your manuscript is now being handed over to our production team.

Kind regards, 

on behalf of

Dr. Meenakshi Thakur 

Academic Editor

PLOS ONE